# An Integrated Perspective on Spatio-Temporal Attention and Infant Language Acquisition

**DOI:** 10.3390/ijerph18041592

**Published:** 2021-02-08

**Authors:** Sofia Russo, Giulia Calignano, Marco Dispaldro, Eloisa Valenza

**Affiliations:** 1Department of Developmental Psychology and Socialization, University of Padua, 35131 Padua, Italy; giulia.calignano@unipd.it (G.C.); eloisa.valenza@unipd.it (E.V.); 2Regionale Beratungs- und Unterstützungszentren (ReBUZ), 28213 Bremen, Germany; mrc.dispaldro@gmail.com

**Keywords:** language acquisition, temporal attention, infancy, development, preverbal, overlap, auditory, longitudinal, speech, syllabic

## Abstract

Efficiency in the early ability to switch attention toward competing visual stimuli (spatial attention) may be linked to future ability to detect rapid acoustic changes in linguistic stimuli (temporal attention). To test this hypothesis, we compared individual performances in the same cohort of Italian-learning infants in two separate tasks: (i) an overlap task, measuring disengagement efficiency for visual stimuli at 4 months (Experiment 1), and (ii) an auditory discrimination task for trochaic syllabic sequences at 7 months (Experiment 2). Our results indicate that an infant’s efficiency in processing competing information in the visual field (i.e., visuospatial attention; Exp. 1) correlates with the subsequent ability to orient temporal attention toward relevant acoustic changes in the speech signal (i.e., temporal attention; Exp. 2). These results point out the involvement of domain-general attentional processes (not specific to language or the sensorial domain) playing a pivotal role in the development of early language skills in infancy.

## 1. Introduction

Ample inter-individual differences exist in the way and rate at which infants learn languages, which may be traced back to the very early stages of cognitive development [1,2]. From the first steps made in language acquisition, infants make use of subtle computational strategies to detect prosodic and statistical cues in the speech stream, and start to identify potential word candidates [3]. Prosodic cues (i.e., stress, rhythm, and intonation) in particular are powerful anchors for the early perceptual system, signaling word boundaries and structure. It is well known that even newborns are sensitive to the acoustic correlations of prosody, also due to their prenatal experience with low-pass filtered speech as perceived in the womb. Due to the low-filter action of the maternal uterine wall (300–400 Hz), speech in the womb loses phonetic details; on the contrary the prosody, melody, and rhythm of languages remain unaltered [4,5]. However, the perceptual and attentional mechanisms (and their interplay) behind the initiation of detecting such prosodic cues are less clear. In particular, here, we investigated the impact of a basic cognitive mechanism of early attentional control (i.e., visual disengagement) on the subsequent ability to process acoustic stress cues (i.e., strong and weak syllables) in the speech stream. The rationale of the present investigation relies on the fact that the development of both language and attention during the first months of life is tightly connected, also due to the gradual emergence of cognitive control on the orienting of attentional focus across domains.

Shortly after birth, indeed, newborns’ attentional system is mainly driven by exogenous stimulation [6]. That is, arousal levels of alertness and readiness are activated by the saliency of the stimuli present in the environment in a bottom-up fashion [7]. From 0 to 2 months of life, the phenomena of “obligatory attention” or “sticky fixation” [8] are indeed common, and efficiency to disengage and shift gaze toward new stimuli only emerges at around 3 to 4 months of life, when the neural maturation of the visual pathway [7,9] and associated subcortical structures [10] allows for the first inhibitory mechanisms and thus, early cortical control. Atypical visual orienting (e.g., longer visual latencies) still present at 7 months has indeed been identified as a prodromal feature of Autism Spectrum Syndrome [11]; moreover, a general “sluggish” ability to disengage attention has been found to be related with developmental language disorders (e.g., Developmental Dyslexia and Specific Language Impairment) [12,13,14,15] and with other neurodevelopmental disorders (including Down syndrome, fragile X syndrome, and William syndrome) [16]. Efficiency in the disengagement of attention is, therefore, an index of early inhibitory control, selective attention, and tendency to explore the external world, with clear positive impacts on communication and language acquisition [17,18].

As in space, the ability to efficiently orient attention in time is crucial in typical language acquisition. The speech stream is, indeed, a complex perceptual stimulus unfolding over time, and attentional resources must be efficiently switched toward different portions of the signal to detect relevant information embedded in the speech stream. To face this challenging task, Astheimer and Sanders [19,20] proposed that Temporally Selective Attention may guide resource allocation toward specific time windows in the speech stream. According to the authors, the time windows selected for further processing are most likely to contain highly relevant information and can be signaled by specific perceptual features, e.g., increased pitch or duration. Consistently, the ability to orient attention over time is supported by a cortico-subcortical network (comprising of the premotor cortex, basal ganglia, and cerebellum), which is already in place since birth [21] and may allow infants to direct attention over time in an exogenous manner (i.e., being attracted by salient prosodic cues such as pitch height, duration, and loudness) [22]. Later in development, due to an increased experience with language, infants narrow their focus of attention to those language-specific features of speech that are predominant in their native language [23,24]. For example, at 7.5 months of age, English-learning infants detect words with a trochaic stress pattern from fluent speech [25] and preferentially direct their attentional resources toward trochaic over iambic syllabic structures at 9 months, being sensitive to the predominant stress pattern of their linguistic environment (i.e., trochees) [23]. Such converging evidence suggests that early language acquisition is shaped by the ability of the attentional system to efficiently orient attentional resources toward rapidly following stimuli, and to preferentially process those portions of the signal that are relevant in the specific environment of infants. In this view, perceptual and attentional mechanisms might work synchronously from the very beginning in shaping the development of successive language skills. Crucially, language–attention interplay measured along the first year of life has been poorly investigated. In particular, we stress the need for longitudinal studies to shed light on individual differences in terms of a mutual attentional and language impact from early infancy [19,22]. The present study moved from this gap to bring preliminary evidence on the influence of early attentional control (i.e., visual disengagement) on the subsequent ability to process language-specific, acoustic cues (i.e., syllabic stress) in early infancy.

### The Present Study

Attentional orienting abilities improve visual perception by intensifying the signal inside the focus of attention and diminishing the noise interference. Given the commonalities between the effect of selective attention mechanisms in the spatial and auditory domains, we hypothesized that the early ability to orient the attentional system in the visual field might predict the successive ability to orient attentional resources toward rapid acoustic changes in the auditory domain, revealing the presence of early basic mechanisms of attention underlying language processing in the first months of life. Moreover, we suggest that a sluggish orienting of automatic attention in the spatial domain could be linked to lower performance in detecting rapid acoustic changes in auditory stimuli. Differences in attention deployment represent a potential candidate as an early marker of atypical development and the present study aimed at providing preliminary evidence on this hypothesis. To verify whether an early efficiency in switching attentional resources toward competing visual stimuli correlates with a subsequent efficiency in detecting rapid acoustic changes in auditory stimuli, we conducted two experiments: (i) in the first experiment, four-month-old infants were tested with an overlap task for visual stimuli with different degrees of visual complexity; (ii) in the second experiment, the same cohort of infants was tested at seven months of age in a discrimination task with linguistic auditory stimuli.

Finally, we explored the link between efficiency of spatial and temporal shifts, correlating individual performances between the two tasks. We hypothesized that an early efficiency in promptly shifting attentional resources toward different visual stimuli (experiment 1) would predict an efficiency in processing prosodic stress cues in the auditory task (experiment 2).

## 2. Experiment 1: Overlap Task

In this experiment, we tested 4-month-old infants with an overlap task measuring the disengagement speed of visual attention. Specifically, attentional disengagement is the ability to interrupt the allocation of cognitive resources deployed toward a given target, allowing the subsequent shift and engagement of attention toward a new object presented in the visual field [26]. The flexibility of this process has been widely investigated with children in early infancy, with the overlap paradigm [27,28]. In this procedure, an infant’s gaze is first drawn to a visual stimulus (S1); then, a second stimulus (S2) is presented in the visual periphery, and this typically draws the eyes from S1 to S2. Saccade latency to reach the second target is an index of the speed of disengagement and can be further computed as a measure of disengagement efficiency. Through the overlap paradigm, it has been found that major development of the disengagement mechanism occurs between 3 and 4 months of age, due to the neural maturation of the posterior attention system—the “where” pathway [7,9]. Age has indeed been found to be an essential factor in developing the ability to shift fixation from a given stimulus to a new one [29], but additional findings also highlighted the role of the stimulus content in the encoding times for S1, and therefore, on the disengagement process [27,30,31]. For instance, Finlay and Ivinskies [31] demonstrated that a comparatively more salient stimulus in the central visual field (S1) makes it more difficult for infants to disengage their gaze. Similarly, Blaga and Colombo [27] demonstrated that 3-month-old infants show slower disengagement for visually complex S1, compared to simple ones. Thus, the speed of processing of a stimulus guides the subsequent allocation of attention toward new upcoming stimuli. In the present study, we combined the overlap paradigm [27,31] with an eye-tracker to detect and compute saccade latency toward the second stimulus (S2), as the dependent measure. In fact, the saccade latency—that is, the time interval between the appearance of S2 and the beginning of the saccadic movement towards it (i.e., the end of fixation of S1)—is an index of speed of disengagement and can be further computed as a measure of disengagement efficiency. Furthermore, we manipulated the information conveyed by S1 in two experimental conditions (simple and complex conditions) to investigate the effects of visual complexity on the overall process [27,31]. We expected to replicate the literature findings, showing that exposure to a more complex content in S1 would produce longer visual processing and, in turn, a slower disengagement latency [27,31]. The disengagement efficiency was also described as the number of trials in which the attentional shift to S2 effectively took place in the two conditions. We expected to observe a lower percentage of correct trials in the complex condition compared to the simple one.

### 2.1. Materials and Methods

#### 2.1.1. Participants

Infants of 4 months of age were recruited by sending letters to families registered in the civil registry; parents who were interested in participating in the study contacted us by telephone or e-mail. The final sample included 15 infants (8 female), of four months of age (mean age = 122 days, SD = 13). This sample is constituted of infants who completed both experiments 1 and 2 of the correlational study; however, these 15 infants are part of a larger cohort of 24 infants who participated only in experiment one. These further data are available in the Appendix A of this paper. The infants (N = 15) included in the present analysis (i) did not drop out of the experimental sessions, and (ii) they completed the whole study (experiments 1 and 2). All of the participants were born and lived in Italy, with both parents speaking Italian as their first language. Exposure to non-native languages was checked and quantified (in terms of hours per week), with no substantial differences identified among the final participant sample. The caregivers of participants were provided with an information sheet before testing and were asked to give further details to check for common risk factors for atypical developmental trajectories (i.e., weeks of gestation, birth weight, sensory disorders and/or familial neurodevelopmental disorders). The collected information showed that the sample does not belong to any atypical or at-risk population. Furthermore, socioeconomic status (SES) indices were investigated for both parents, as they have a potential impact on infants’ development, including the number of family members, year of birth, level of education, occupation, language, and nationality [32,33]. From these reports, all the families participating in the study were homogeneously ranked as having a high-level SES. The whole research protocol was approved by the departmental ethics committee and conducted in accordance with the principles laid down in the Declaration of Helsinki. Parents provided written informed consent.

#### 2.1.2. Apparatus

Data collection was implemented by using the video-oculography (VOG) technique to record eye movements. The eye-tracker (Tobii X2-60, Tobii Pro AB, Stockholm, Sweden) was placed below a 27-inch led screen (Philips, Eindhoven, Netherlands, E Line Blacklight Monitor, 300 × 300,). The monitor and the eye-tracking camera were connected to a laptop (Acer travel mate 5772 g, Acer Inc., New Taipei, Taiwan) using E-Prime 2.0 to present visual and audio stimuli. A dark curtain in the room separated the researcher’s area from the area occupied by the participant. A second camera placed onto the screen made it possible to look at participants performing the test. The audio stimuli were presented with two speakers (KRK RP5 RoKit G3, KRK Systems, Deerfield Beach, FL, United States) placed on the right and left of the monitor. We used a dimmer to lower the brightness of the room and obtain a constant luminance, setting proper signal detection to be provided for the eye-tracker.

#### 2.1.3. Stimuli

Visual stimuli were squares of different dimensions and internal patterns. The pattern of the squares included red and white lines vertically or horizontally oriented, or a red and white checkerboard. Presentation of the central stimulus varied according to two different conditions: In the simple condition (S), the internal pattern of the central stimulus remained the same while the dimension changed, alternating smaller squares (8 × 8 cm^2^) and bigger ones (10 × 10 cm^2^), so it appeared as a flickering image (Figure 1a). In the complex condition (C), both the internal patterns and the dimensions of the central stimulus changed every 500 milliseconds, resulting in a perceptually more complex image compared to that presented in the simple condition (Figure 1b). The peripheral stimulus was a static, monochromatic square, and it measured 10 × 10 cm^2^. It could appear at the left or right of the central stimulus, in random order. The areas of the images corresponded precisely to the areas of interest, measuring 10 × 10 cm^2^.

#### 2.1.4. Procedure

On their arrival, families were welcomed in the waiting room. After signing the consent form, caregivers and infants were conducted to the research laboratory. There, we first let infants familiarize themselves with the lab setting, and then the participants were set in an age-appropriate padded chair in front of a computer monitor, with their caregivers behind. An introductory video engaged the attention of the infants, enabling the eye-tracker to find the corneal reflex and trace their eye movements. A 5-point calibration procedure with attention-getting, audiovisual targets was initially conducted. Five colored cartoons, each accompanied by an engaging soundtrack, were presented one by one and occupied the marker positions corresponding to the areas of interest: the top left, top right, center, bottom left, and bottom right. Following a successful calibration, the task began. At the beginning of each test trial, the central stimulus (S1) appeared on the screen, providing the first area of interest for the eye-tracker (AOI; 10 × 10 cm^2^). Once the infant reached 1500 ms of fixation on S1, a second stimulus was presented, providing the second area of interest (AOI; 10 × 10 cm^2^). The second, peripheral stimulus (S2) was a static square, and it could appear to the left or right of S1; S1 continued to be present even after S2 appeared (attentional overlap paradigm). Generally, when the second stimulus appeared on the screen, infants interrupted their fixation on S1 (disengagement), made a saccade from the center to the peripheral stimulus (shifting), and started a new fixation on S2 (engagement); when the infants accumulated 200 ms of fixation on S2, then the trial ended. When a saccade was not registered after 5000 ms from the appearance of the peripheral stimulus, the trial was interrupted, and a new one started. An attention grabber appeared at the center of the screen after each stimulus presentation, allowing the infant’s gaze to be centered at the beginning of each trial. The test included 24 trials, 12 for the complex and 12 for the simple condition, presented in a fully randomized order. After presentation of the first 12 trials, a cartoon video was presented to sustain the infant’s attention. As the test proceeded, we asked the parents not to interact with the infant unless necessary; in this second case, the test was interrupted and restarted only if the infant felt comfortable again. The whole procedure lasted around 5 min.

### 2.2. Results

We conducted analyses on a sample of 15 infants (8 females; mean age = 122 days, SD = 13). Inclusion criteria for participants were set before data collection and consisted of: (a) a minimum of three valid trials per condition, and (b) to have completed both experiments 1 and 2 of the study. For a trial to be coded as valid, the infant had to: (a) successfully accumulate 1500 ms of looking time on S1, (b) be fixating on S1 at the point when the peripheral stimulus (S2) appeared, (c) make an overt shift in the direction of the peripheral stimulus, and (d) accumulate 200 ms of looking time on S2. The dependent measure was the disengagement latency of the infants, defined as the time interval between the appearance of S2 and the first saccadic movement coming out of S1, recorded automatically and continuously by the eye-tracker. Figure 2 (left panel) shows the descriptive statistics for the latency times during the simple and complex condition, respectively.

#### 2.2.1. Statistical Analysis

Data were analyzed with the freely available, open-source R software [34]. Outliers were evaluated by the influence analysis for generalized mixed-effects models [35]. Note that GzLMs are an extension of the GLMs that allow specification of the distribution family and the random effects, i.e., individual variability. When needed, this overcomes the assumptions, made by ANOVAs and GLMs, that residuals should be normally distributed, and their variability should be uniform across the levels of the predictors [36]. All models were fitted with the lme4 package [37]. To find the best approximation to the true model, we followed a model comparison approach, using the likelihood ratio test (LRT), Akaike Information Criterion (AIC); [38] and AIC weight as indexes of the goodness of fit, with the former testing the hypothesis of no differences between the likelihoods of two nested models. The AIC and AIC weight give information on the relative evidence of models (i.e., likelihood and parsimony) so that the model with the lowest AIC and the highest AIC weight is to be preferred [39].

#### 2.2.2. Disengagement Latency

The average disengagement latency was 605 ms (SD = 329) in the simple condition (S), and 931 ms (SD = 833) in the complex one (C). Table 1 shows that the complex condition predicted longer disengagement latency compared to the simple one (b = 0.51, SE = 0.18, t = 2.87). These results suggest that the content of the midline stimulus affected ocular latencies: the more complex the content of the central stimulus, the more time to disengage emerged. Figure 2 (right panel) shows the percentage of trials in which no attentional disengagement was observed (i.e., no attentional shift from S1 to S2) in the simple and complex conditions, respectively. The ratio was computed on the total performed trials completed by each participant.

As can be seen, there is higher inter-individual variability in the complex condition than the simple one: indeed, a significant number of participants did not show any gaze disengagement in the complex condition compared to the simple condition (χ = 5.9, df = 1, *p*-value = 0.01). Table 2 shows that the number of trials during which infants did not exhibit any disengagement of attention was substantially different between the two conditions: more trials with no gaze disengagement were registered on the complex condition than the simple condition (b = 0.6, SE = 0.21, t = 2.92).

Overall, the results of Experiment 1 suggest the presence of attentional disengagement abilities in four-month-old infants; moreover, a significant influence of the central stimulus content was registered: S1 complexity influenced both the latency and the occurrence of infant disengagement.

### 2.3. Discussion

Experiment 1 explored the disengagement ability of 4-month-old infants, focusing on the effect of stimulus complexity on selective spatial attention when competing for information—that is, when two visual stimuli were simultaneously presented (overlap task). Our findings replicate those of Blaga and Colombo [27], showing that the speed of disengagement at 4 months depends on the visual complexity of S1: a comparatively more salient content on S1 indeed slowed down the saccade latency during attentional disengagement, compared with the simplest. Moreover, the results showed that the simple condition increased the likelihood of infants disengaging their attention and producing a saccade toward the peripheral stimulus, compared to the complex condition. Such a pattern of results suggests that infants who are able to rapidly disengage attention according to the perceptual saliency of the stimulus are more likely to relocate their attentional focus towards relevant information in space. In line with previous findings, this result is generally thought to reflect the maturation of the so-called “posterior attention system” or “where” pathway, e.g., [40], identified as the neural substrate for regulating the orienting of attention. In particular, the ability to disengage attention seems to be mediated by the posterior parietal lobe [29] and major improvements in this ability appear to occur precisely between 3 and 4 months of age, with moderate changes observed after this first maturational window [28]. Therefore, visual orienting mechanisms are already in place at four months of age, despite being significantly influenced by exogenous factors. In Exp. 1, we explored the effect of stimulus complexity, finding that a comparatively more complex content in S1 slowed down disengagement latencies and reduced disengagement occurrence. In line with the processing hypothesis of Blaga and Colombo [27], we interpreted such findings by considering the time needed by each infant to encode and process more complex visual content. By focusing on the strict link between look durations (considered as a measure of encoding speed) and disengagement abilities at four months [28], a more complex content in S1 might request more attentional resources to be processed, slowing down the disengagement latencies or even disrupting the entire processing of the peripheral stimulus.

Furthermore, the results presented here are corroborated by those of the larger sample to which they belong (N = 24), presented in the Appendix A. Therefore, experiment 1 showed that: (i) infants at 4 months are able to process competing information in the visual field, (ii) an efficient disengagement is essential to orient attention toward different stimuli, and (iii) stimulus complexity strongly influences the saccade latency and disengagement occurrence. Starting from these findings on spatial attentional shifting, we proceeded by examining how temporal attentional resources are deployed toward changes in linguistic, auditory stimuli presented through time, in the same cohort of infants.

## 3. Experiment 2: Discrimination Task

The attentional system is strongly time-dependent by competing for auditory information. Indeed, to identify and select auditory stimuli, as it occurs in language, a rapid shift of attention along time (i.e., temporal attention) is fundamental in order to detect segmental information embedded in the speech stream. Precisely, temporal attention implies identifying and selecting specific points in time for further processing [19,20]. For instance, an efficient allocation of temporal attention is required to detect brief syllabic variations in terms of frequency, intensity, and duration (i.e., prosodic cues).

Prosodic cues are prominent in natural speech and help infants to identify potential word candidates [3]. Considering the stress dimension, for example, about 90% of English multisyllabic words begin with linguistic stress on the first syllable, as in the words “pencil” [pɛnsəl] and “stapler” [steɪplər] [41]. This strong–weak (trochaic) pattern is the opposite of that used in other languages—for instance, in Polish, a weak–strong (iambic) pattern predominates. In fact, all languages contain words with both kinds of stress pattern, but one pattern typically predominates over the other. At 7.5 months of age, English-learning infants segment words from speech that show a strong–weak pattern, but not those that show a weak–strong pattern; moreover, they tend to treat strong syllables as word-onsets. Indeed, when infants hear “guitar is”, they perceive “taris” as a word-like unit [25,42]. This scenario also belongs to our participants’ mother tongue (i.e., Italian) because Italian words more frequently present lexical stress on the first or penultimate syllable [43,44]. Hence, the prosodic structure of continuous speech constrains the mechanisms relating to the orienting of attention and biases word segmentation towards familiar patterns across languages. Nevertheless, the attentional behavior toward this prosodic cue has been less investigated among Italian-learning infants.

Given the temporal nature of speech, here, we aimed at investigating how temporal attentional resources are deployed to process rapid acoustic changes occurring in prosodic stress patterns. Specifically, in Experiment 2, we investigated whether 7-month-old, Italian-learning infants were able to discriminate change across trochaic syllabic sequences as a relevant prosodic cue in their native language. To do so, we first familiarized infants with strong–weak syllabic sequences, and then we manipulated this pattern by substituting the strong or the weak syllable in a subsequent test phase with a pure tone of the same duration, pitch, and loudness as the replaced syllable. By doing so, we explored the effect of high salient (i.e., strong syllable), versus low salient (i.e., weak syllable), acoustic cues on attentional resource allocation and on change detection abilities occurring in syllabic sequences. We hypothesized that a preferential resource allocation would be directed toward the most relevant perceptual cue, and so the duration and saliency of the strong, but not the weak, syllable should trigger attentional resources in detecting changes in the overall syllabic sequence. Therefore, we expected changes in the syllabic pattern to be detected in the case of strong, but not weak, substitution.

### 3.1. Materials and Methods

#### 3.1.1. Participants

The fifteen Italian-learning infants (8 female) who completed the Experiment 1 also took part in Experiment 2 at seven months of age (mean age = 218 days, SD = 22). Parents were contacted by telephone or e-mail to plan a second appointment, and they provided a written informed consent also for this second part of the study.

#### 3.1.2. Apparatus

As in Experiment 1, data collection was carried out using the video-oculography (VOG) technique to record participants’ eye movements, with an eye-tracker (Tobii X2-60). See experiment 1 for more details.

#### 3.1.3. Stimuli

The stimuli consisted of audio and visual items. The audio stimuli were sequences of disyllables spoken by a female voice. Each sound sequence had a consonant–vowel (CV) structure, with a strong–weak (Sw) syllabic stress pattern. Ten consonants were selected: 6 occlusive (/b/, /t/, /k/, /p/, /g/, /d/), 2 nasal (/m/, /n/), 1 fricative (/v/), and 1 lateral (/l/); and three vowels (/a/, /o/, /u/). The vowels /i/ and /e/ were not included because they have been found to introduce a confounder in that they are harder to distinguish [45]. The audio stimuli were presented in a loop, with a 400 ms pause before each sequence (i.e., pause—NAnaNAna—pause). A trochaic pattern consisted of a sequence with the main stress on the first syllable, which is typical in the mother tongue of the participants [43,44]. Crucially, the CV structure was changed from one trial to another to exclude any familiarization effect due to the repetition of the same phonemic sequence. Moreover, CVCV sequences were balanced across conditions for duration, dB, and Hz, and also for phonetic features. Indeed, each condition displayed stimuli from each phonetic category (i.e., occlusive, nasal, fricative, and lateral). The same consonant could recur in different trials, but it was matched each time with a different vowel; each trial thus involved a different syllabic pattern. A pure tone, with the same features as the syllable being replaced (in terms of milliseconds, dB, and Hz), was used to manipulate the sequences by alternately substituting the strong (i.e., Sw#w, e.g., NAna#na) or the weak syllable (i.e., SwS#, e.g., NAnaNA#). Table 3 shows the mean acoustic features of stimuli across the test conditions. All the acoustic stimuli were registered, analyzed, and edited with the software PRAAT [46].

The visual stimuli consisted of twenty-four static and anthropomorphic cartoons, appearing on a grey background at 9.554 deg (10 × 10 cm^2^). The areas of the images exactly corresponded to the areas of interest (AOI) and measured 10 × 10 cm^2^ (9.554 deg, 60 cm away from the display). An image was randomly matched with any soundtrack and remained visible throughout the entire trial. The visual stimuli (corresponding to the AOI) could appear in 5 different positions on the screen, in order to catch the infant’s attention.

#### 3.1.4. Procedure

Testing began after familiarizing the infants with the lab setting and after eye-tracker calibration (see experiment 1 to more details). Participants were tested by a familiarization preference procedure [47] readapted with trochaic syllabic stimuli and an eye-tracker system (Figure 3) [48,49]. The familiarization phase (FP) consisted of a static cartoon image presented on the screen. After the participant had fixated the visual image for 300 ms, the soundtrack automatically started and continued up to a maximum of 15 s. When the infant’s gaze moved away from the image for more than 200 ms, the soundtrack stopped, and the visual stimulus moved to another AOI to trigger the infant’s attention again. If the infant looked away for more than 2 s, the trial was considered invalid. The FP was considered complete once the infant heard the soundtrack for at least 8 out of 15 s. The test phase (TP) started immediately after the FP (Figure 3a) and included three different conditions (Figure 3b), each composed of 8 trials. The visual image and the CV sequence of each TP trial were the same as in the corresponding FP trial, but the syllabic sequence was changed according to three experimental conditions: in the Familiar condition (F), the syllabic pattern was the same as in the FP (i.e., NAnaNAna); in the Novel Strong condition (NS), the syllabic pattern was changed by substituting the strong syllable (i.e., Sw#w, e.g., NAna#na); and in the Novel Weak condition (NW), the syllabic pattern was changed by substituting the weak syllable (i.e., SwS#, e.g., NAnaNA#). In the NS and the NW conditions, the substituted syllable was replaced by a pure tone with the same mean duration (in milliseconds), dB, and Hz (*p* > 0.05) as the syllables it substituted. The substituted syllables were also comparable (in terms of frequency, duration, and decibels) within the 3 conditions. The 24 trials in total composing the test phase (8 trials per condition: F, NS, and NW) were randomly presented and divided into two blocks by a mute cartoon video in order to maintain the infant’s attention. The random presentation was made in order to exclude possible predictability in the TP sound sequence.

For all the TP conditions, the soundtrack only started after 300 ms of fixation on the new visual stimulus; hence, only fixations directed towards the visual stimulus while the soundtrack was playing were considered as valid. Each trial lasted no more than 6 s. When the infant’s gaze moved away from the image for more than 200 ms, the soundtrack stopped, and the visual stimulus moved to another AOI to trigger the infant’s attention again. If the infant looked away for more than 2 s, the trial was considered invalid. A trial was considered complete only once the infant heard the soundtrack for at least 2 of the 6 s. After each trial, an intra-stimulus interval (ISI) consisting of a black screen began and lasted for 100 ms. All of the infants were tested under each of the 3 TP conditions (i.e., 24 trials, 8 per condition) and were included in the analyses only if they reached a minimum of 2 valid trials per condition. The whole procedure lasted around 5 min.

### 3.2. Results

#### Looking Times

Figure 4 shows the descriptive statistics for looking time in milliseconds, in each of the 3 conditions (F, NS and NW) of the test phase (TP). Infant looking time was longer in the Novel Strong condition (NS mean = 5149 ms, SD = 695 ms) compared with the Novel Weak condition (NW mean = 4507 ms, SD = 1095 ms), which was longer than that registered in the Familiar condition (F mean = 4324 ms, SD = 953 ms). The model comparison (see Table 4) showed that looking time was significantly longer in the NS condition compared to both the NW (b = 808, SE = 221, t = 3.656) and the F (b = 931.3, SE = 218, t = 4.27) conditions, whereas no difference emerged between the NW and the F conditions (b = 123.5, SE = 196, t = 0.63).

These results indicate that the participants were able to discriminate auditory changes within trochaic syllabic sequences only when the strong, but not the weak, syllable was substituted. This finding suggests a preferential processing of strong syllables, which demand less resource allocation, over weak ones.

A percentage of listening times for each condition was computed as the ratio of the time spent on a given condition to the total time reached by each participant across conditions (e.g., NS = NS/(F + NS + NW)). Then, we calculated an individual, auditory discrimination index, computing the difference between the listening time percentage in the NS and NW conditions and the listening time percentage for the F condition. The computation of the auditory discrimination index allowed us to sharply analyze performance at the individual level (see Figure 5): indeed, a discrimination index less than or equal to zero reveals that the infant listened for the same amount of time to familiar and novel sequences and, consequently, probably had not detected the auditory change. On the contrary, a discrimination index greater than zero indicates that the infant listened longer to the manipulated sequences and, as a consequence, probably detected the acoustic variation in the sequence.

### 3.3. Discussion

Experiment 2 explored the acoustic discrimination ability of 7-month-old infants, focusing on the effect of stimulus saliency on the deployment of temporally selective attention toward trochaic syllabic sequences. The results show that the infants were able to detect the auditory change occurring with strong, but not weak, syllables in previously familiarized sequences, compared to a control condition. Accordingly, the fixation times show that the infants spent more looking time at images while hearing a novel auditory pattern, but only if the syllabic change was highly perceptible (i.e., strong syllable replaced; NS) and not if it was scarcely perceptible (i.e., weak syllable replaced; NW). Our results suggest that the high saliency (driven by the higher duration, loudness, and pitch height) of strong syllables triggers greater resource allocation during auditory change than weak syllables. Complementary to this, the demanding information processing required by the low salient features of the weak syllables reduces the chance of any change being noticed. Overall, 7-month-old Italian learning infants were found to detect acoustic change only when it occurred with strong, but not weak, syllables within trochaic sequences. The processing of strong–weak (trochaic) stress patterns for English-learning infants is well documented: for example, it is known that at 7.5 months of age, infants are familiar with the predominant stress pattern of their native language, being able to identify trochaic words from continuous speech and to treat strong syllables as cues to word onset [25,42]. This early sensitivity to stress cues is an adaptive strategy since word-onsets have been found to be less predictable and more informative for word recognition than medial and final segments [50,51,52]. Astheimer and Sanders [19,20] interpreted preferential processing of word-onsets as the influence of temporal selective attention in guiding resource allocation toward relevant time windows in the speech stream. Similarly, trochaic patterns predominate in the native language of our participants (i.e., Italian) [43,44] and, mirroring the finding from English-learning infants, the ability to recognize words and syllabic patterns in Italian-learning infants has been found to depend more on strong syllables, and less on weak syllables at 7 and 11 months [48,53]. Therefore, an infant bias toward the prosodic stress cues typical of their target language has been widely tracked across languages. In Experiment 2, we investigated the attentional mechanisms underlying the ability of Italian infants to detect changes in informative prosodic cues, in order to discriminate an auditory change at the segmental level. Instead of real words, we employed CVCV syllabic sequences since we reasoned that early perceptual and attentive mechanisms are already in place in the processing of speech sounds, even before any semantic knowledge is activated in the speech stimulus. Indeed, we wanted to analyze attentional deployment toward high salient (i.e., strong syllables) versus low salient (i.e., weak syllables) acoustic changes occurring within rapid syllabic sounds.

Overall, experiment 2 showed that: (i) Italian infants at 7 months of age detect acoustic changes occurring in familiar syllabic sequences; (ii) the attentional system preferentially orients attentional resources toward specific time windows, signaled through high salient perceptual features; and (iii) this process may have a side effect of leaving those changes occurring with less perceivable linguistic segments unnoticed. The further step we navigated was exploring the link between the spatial orienting of attention at four months and the subsequent language processing skills registered at seven months.

## 4. Correlational Analyses

To investigate the relation between inter-individual variations in spatial attention (measured as disengagement efficiency at four months; Exp. 1) and auditory temporal attention for linguistic stimuli (measured as rapid syllabic changes detection at seven months; Exp. 2), we correlated the percentage of failed trials, (i.e., the percentage of trials with no disengagement registered), with the discrimination index (i.e., the percentage of time spent on novel acoustic sequences) by using Spearman’s rank correlation.

We expected to observe a negative correlation between the percentage of failed trials registered in the disengagement paradigm and the percentage of listening time for the novel sequences in the acoustic discrimination task; that is, infants showing a higher difficulty in disengaging attention from visual stimuli might also exhibit a reduced ability in detecting acoustic changes occurring in linguistic stimuli. As shown in Figure 6, correlational analyses performed between the two experiments (Exp. 1 and Exp. 2) showed that infants exhibiting a reduced ability to discriminate acoustic variations on strong syllables at 7 months already showed a higher difficulty in disengaging attention from a perceptually simple stimulus in the visual field at 4 months. Indeed, a substantial negative correlation emerged between the failed trials percentage in the simple disengagement condition and the discrimination index in the NS condition.

## 5. General Discussion

The ability to efficiently orient attentional resources toward perceptually composite stimuli has long been studied within the field of cognitive development, both in infancy and early childhood. Regarding language development, a sluggish orienting of attention has been found to substantially impact language acquisition, as documented by different studies on atypical development. In particular, children and adolescents with Specific Language Impairments and Developmental Dyslexia have been found to struggle in detecting short series of both auditory and visual stimuli, when the amount of available time is limited [12,13,14,15]. Similarly, visual disengagement performances (i.e., speed of disengagement) are associated with atypical language acquisition in infants and toddlers with several chromosomic clinical pictures (i.e., Down syndrome, William syndrome, Fragile X syndrome) [16], and Autism Spectrum Disorder [54].

Given the substantial effect of spatial–temporal attention on atypical language development, it becomes crucial to understand what role these underlying cognitive mechanisms play from the very early stages of language acquisition. To investigate this issue, in the current study, we analyzed the attentional mechanisms underlying early visuospatial (Exp. 1) and auditory (Exp. 2) processing in typically developing infants. Specifically, we examined whether inter-individual performances on visual attentional switching at four months were related to speech discrimination ability at seven months. Both the overlap task (Exp. 1) and the auditory-discrimination task (Exp. 2) emerged as suitable paradigms to measure the automatic deployment of attentional resources in response to changes in the visual and auditory environment (spatial and temporal attention; [19,20,27]). Through these two tasks, we showed that 7-month-old infants who exhibited low efficiency in discriminating a rapid auditory change already exhibited low efficiency in visual attentional disengagement at four months. Indeed, a low discrimination index in Exp. 2 significantly correlated with a higher percentage of failed trials in Exp. 1. Notably, results concern a sample of typically developing infants, who were included in the research since they were not considered at-risk of developmental disorders. Nevertheless, a strong inter-participant variability was registered, which allowed us to focus on the study of individual differences. Therefore, the computation of the auditory discrimination index and the percentage of failed trials allowed us to analyze performances at the individual level, which was the main focus of the study. However, the high inter-participant variability emerging from the results, besides being qualitatively interpreted, must be further discussed in terms of sample size and statistical power. Indeed, despite being comparable to similar studies in the field (especially regarding longitudinal research), our sample size was small (n = 15), and therefore, further research is needed to gain substantial statistical power. Nevertheless, in our study, we took advantage of generalized mixed-effects regression models performed on both exp. 1 and exp. 2. This approach is indeed a suitable statistical technique for dealing with both within- and between-participant variance. Importantly, the effects estimated at the group level consider individual variability, hence offering more plausible effects than traditional approaches, e.g., ANOVA. Thus, we maximized the likelihood of capturing plausible effects at the subject and the group level. Therefore, even if the evidence we found should be considered preliminary and further replications are needed, some preliminary conclusions can be drawn.

In particular, our results seem to indicate that the extent to which infants exhibit difficulties in processing competing information in the visual field may have a parallel with their subsequent ability to orient temporal attention toward relevant time windows in the speech signal. Indeed, in front of a general tendency of the attentional system to preferentially focus on the most salient stimulus present in the visual (i.e., complex S1) and auditory (i.e., strong syllable) fields, we found a substantial correlation between individual performances in the simple conditions of exp. 1 (i.e., simple S1) and exp. 2 (i.e., NS). Therefore, it seems that the two simple conditions offered an excellent experimental constraint that led to individual differences emerging, whereas in both the complex conditions (i.e., complex S1 in exp. 1 and NW condition in exp. 2), a floor effect was registered since a significative part of subjects failed to perform the trials. In conclusion, the simple conditions of both exp. 1 (i.e., simple S1) and exp. 2 (i.e., NS) were useful for letting the baseline attentional profile emerge. Instead, the complex conditions of both exp. 1 (i.e., complex S1) and exp. 2 (i.e., NW) were useful to investigate the impact of stimulus content on attentional engagement and, therefore, on the ability to disengage.

The ability to efficiently disengage from one stimulus and shift attention toward another can indeed be further interpreted in light of encoding speed and attentional engagement, on which the stimulus content had a considerable impact. Indeed, in order to reach good performances both in the overlap and in the auditory discrimination task, participants had to rapidly process visual (Exp. 1) and auditory (Exp. 2) information. Data from Experiment 1 show that the content of the stimulus that has to be processed (S1) influences the encoding time: when S1 is enriched with visual information (i.e., in the complex condition), more time is demanded to encode it, and, as a consequence, longer time to disengage is requested. This explanation is in line with the processing hypothesis proposed by Blaga and Colombo [27], who suggest that at least some of the age-related change in infant performance on the overlap paradigm may be attributable to the speed of processing midline stimulus. In the same vein, data of Experiment 2 suggested that the weak syllable requires more processing speed to be processed than the strong syllable: when infants do not have enough time to encode a syllable, as probably occurred for the weak syllable, they are not able to detect the variation in the tone pattern. Our explanation is in line with previous data collected by Hary and Renvall [12], showing that individuals who are slower at directing attentional resources towards an auditory target are also slower at directing attentional resources on a visual target. Moreover, those individuals are less likely to succeed in tasks involving the ability to encode competing visual or auditory information simultaneously, with significant impacts on language processing [12]. Indeed, a sluggish attentional orienting has mainly been reported in studies investigating children with language impairments [12,13,14,15].

A further interpretation (not in contrast with the previous one) can be given considering the gradual emerging of cognitive control over attention deployment in infancy. Indeed, early in life, attention is mainly driven by exogenous stimulation [6], with the intrinsic saliency of stimuli being highly attractive for the developing system. Given the limited amount of attentional resources, preferential focusing on salient stimuli let the early attentional system orient through the overwhelming environment surrounding the infant by acting as a filter on the incoming stimulation. Therefore, the selective processing of highly salient stimuli in the environment, rather than being considered a disadvantage, allows the incoming information to be further processed by operating a simplification for the system, as suggested by the Less is more hypothesis [22,55]. Consistently, our results show that, at 4 months of age, an early disengagement ability is influenced by the midline stimulus content, with major resource allocation displayed toward complex vs. simple stimuli. Similarly, a tone-change occurring over strong, but not weak, syllables was discriminated by 7-month-old infants, with the higher saliency of strong syllables (in terms of duration, loudness, and pitch heigh) probably aiding this process. Besides their intrinsic saliency, strong syllables might have been preferentially attended to also due to the considerable amount of experience with their native language of 7-months-aged infants. Indeed, infants participating in our study were all born and lived in Italy, with both parents speaking Italian as their first language. As in the case of English or Dutch, the majority of words in the Italian language present a trochaic stress pattern, which is characterized by a strong/weak syllabic structure [43,44]. Thus, familiarity with the stress structure of CVCV sequences might have driven the preferential resource allocation toward the strong syllables, representing an anchor for the attentional system. This strategy might be particularly adaptive since word onsets are less predictable and more informative for word recognition than medial and final segments [50,51,52]. Therefore, at 7 months of age, the infants’ focus of attention is also influenced by the familiarity with the stimuli, besides their intrinsic saliency.

In conclusion, our findings indicate that: (i) early spatial and temporal mechanisms drive the attentional system by biasing attentional resources toward relevant portions of crowded (visual and auditory) signals, and (ii) this tight coupling between attentive and perceptual systems has an impact on early speech processing. Indeed, by testing the same sample over time, we described how infants showing a reduced ability to discriminate acoustic variations across syllables at seven months already showed a higher difficulty to disengage attention from a simple visual stimulus at four months. This might be explained by the critical role played by early attentional and perceptual mechanisms acting across different sensorial modalities and from the very early stages of cognitive development, in sustaining successive language acquisition. Our results indicate that a rapid shift of visual attention (related to an efficient disengagement ability) might predict the future emergence and development of language processing skills. Therefore, these findings offer preliminary and fruitful insights on the role played by spatial and temporal attention in typical language development. Those results also help shed light on the documented link between the orienting of attention and language acquisition across developmental disorders. Indeed, despite our participants belonging to a typically developing sample, the huge inter-participant variability registered confirms that attentional development can follow different possible trajectories, also requiring qualitative approaches to be understood. Furthermore, our results demonstrate that variability in language outcomes can be related to variability in early visual disengagement even in a sample of typically developing infants, as also reported from a recent study by D’Souza and colleagues [16]. In particular, the research from D’Souza and colleagues (2020) reported that performances in visual disengagement of attention are associated with subsequent language development in both typically developing infants and other chromosomic clinical profiles (i.e., Down syndrome, William syndrome, and Fragile X syndrome). This evidence adds up with the already discussed research on developing language disorders (i.e., Developmental Dyslexia and Specific Language Impairments). The resulting picture points to a relevant effect of attentional orienting in shaping language acquisition already from early infancy. Further research is needed to clarify the variegated scenario of infants’ attentional behaviors and to interpret the nature of individual performances and inter-participant variability, in the atypical as in the typical development. This study contributed to this challenging purpose by bringing preliminary evidence on the tight link between attentional, visual orienting, and auditory stress perception in early infancy.

## 6. Conclusions

The present study revealed that rapid and efficient information encoding, especially of those stimuli enriched with information (complex visual stimuli and stressed syllables), impacts both visual disengagement and auditory discrimination ability in the first year of life. The present investigation, focusing on individual differences, indicates that those infants that slow down information encoding and were not influenced by complex visual feature during disengagement of attention (by treating simple and complex stimuli as similar), at 4 months of age, did not discriminate any auditory change occurring in a familiar syllabic pattern, at 7 months. We show that infants who slow down the encoding of visual and auditory stimuli and that do not prioritize salient visual and auditory features were less effective in both visual attentional disengagement and auditory discrimination. The proposed attentional tasks are suitable candidate tools to measure the efficiency of the spatial–temporal attentional mechanisms under the condition of fast encoding, along developmental stages. Despite these results represents only preliminary evidence and further research is needed, they contribute to the debate on early markers identification and prevention strategies of developmental language disorders. By capitalizing on efficient experimental paradigms, we focused on individual differences in attentional deployment and language acquisition. As the Neuroconstructivist view of development suggested [56], development itself is the key to understanding developmental disorders. Accordingly, we strongly encourage further studies to explore the roots of inter-individual differences in domain-general mechanisms involved (also) in language acquisition from early infancy and over time.

## Figures and Tables

**Figure 1 ijerph-18-01592-f001:**
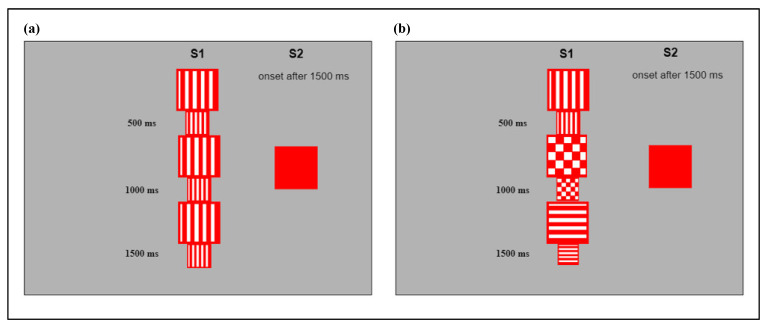
(**a**) Example of the central (S1) and peripheral (S2) stimuli presented during the simple condition; (**b**) Example of the central (S1) and peripheral (S2) stimuli presented during the complex condition.

**Figure 2 ijerph-18-01592-f002:**
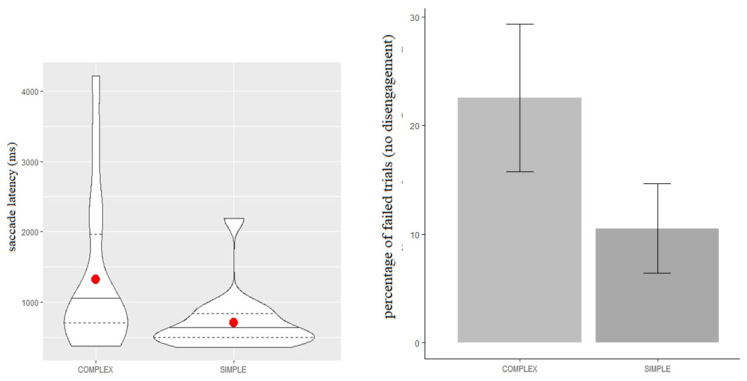
Violin plot of disengagement latency (left panel) and bar plot of percentage of failed trials and associated standard errors (right panel) for the complex and simple condition. Median (red dot), first and last quartile (dashed line) of data distribution are drawn.

**Figure 3 ijerph-18-01592-f003:**
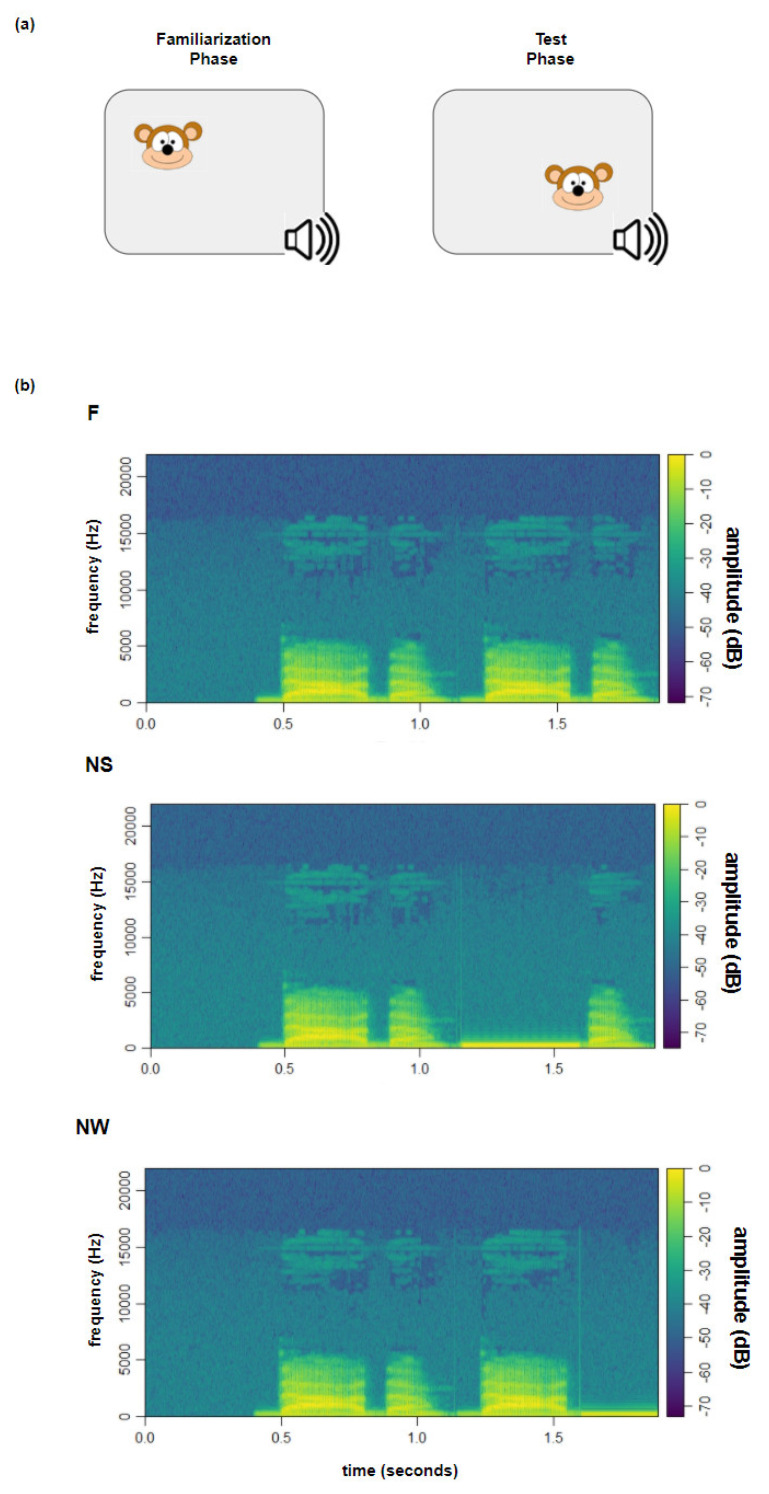
(**a**) Familiarization task, consisting of the familiarization (FP) and the test (TP) phase; and (**b**) spectrogram of frequencies (left vertical axis) and an amplitude spectrum (right vertical axis) of the auditory sequence during the familiar phase, e.g., /Dada/ /Dada/ of audio stimuli across conditions for the test phase (i.e., F, NS, NW). Note that the stimuli of the familiarization phase are the same as those presented during the familiar (F) condition in the test phase.

**Figure 4 ijerph-18-01592-f004:**
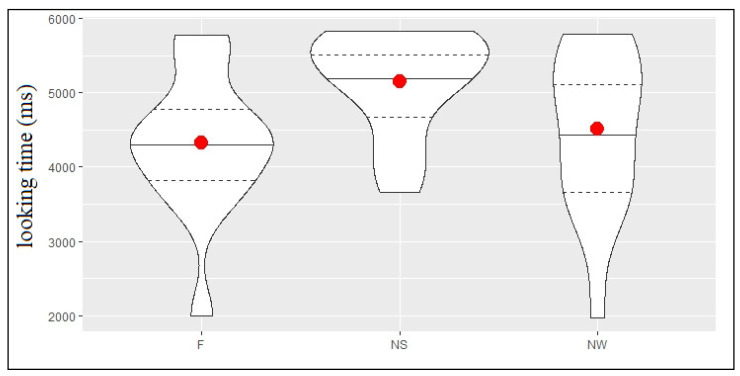
Violin plot for looking time in milliseconds during the familiar (F), novel strong (NS), and novel weak (NW) condition. Median, first and third quartile, and outliers are plotted.

**Figure 5 ijerph-18-01592-f005:**
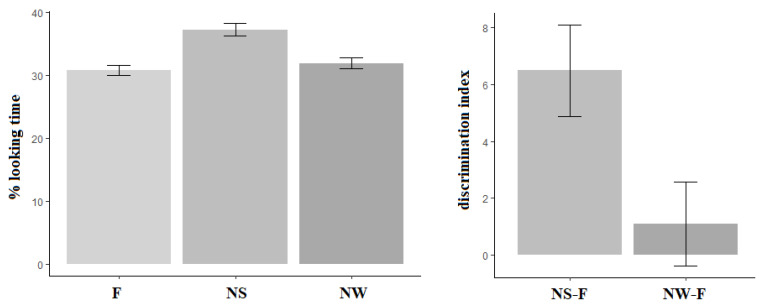
Bar plots in the left panel: the percentage of listening time and associated standard error during the discrimination task conditions, computed as the ratio between the total looking time of each condition and the time spent on the overall conditions by individuals (e.g., F = F/(F+ NS +NW)). In the right panel: the discrimination index and associated standard error computed as the difference between the percentage of listening times during NS, NW, and F. Values > 0 indicate longer times on the NS and NW conditions, respectively.

**Figure 6 ijerph-18-01592-f006:**
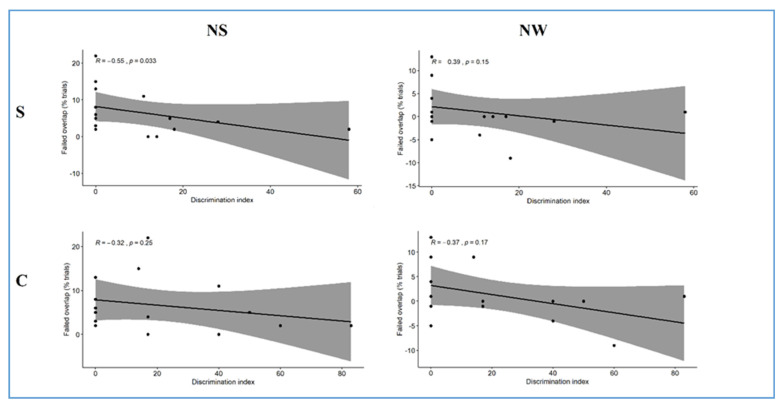
Spearman correlation plot of the percentage of trials without disengagement of attention in the simple (S) and complex (C) overlap conditions at 4 months, and the discrimination index during the novel strong (NS) and novel weak (NW) conditions on the syllabic discrimination task at 7 months. R = Spearman’s rank correlation coefficient.

**Table 1 ijerph-18-01592-t001:** Model comparison for predicting disengagement latency in the overlap task.

Model	RD	dAIC	AICw	df	χ	*p*	η^2^
Saccade latency ~ (1|participant)	27	6.8	0.03	\	\	\	\
Saccade latency ~ condition + (1|participant)	26	0	0.97	1	8.78	0.003	0.02

RD = residual deviance, AIC = Akaike information criterion, dAIC = difference between a model’s AIC and those of the best model, AICw = AIC weight, df = degrees of freedom of the chi-squared statistic, η^2^ = eta squared as the ratio between the chi-squared statistic and the residual deviance of the null model.

**Table 2 ijerph-18-01592-t002:** Model comparison for predicting the number of trials with no disengagement in the overlap task.

Model	RD	dAIC	AICw	df	χ	*p*	η^2^
No disengagement ~ (1|participant)	13	6.2	0.04	\	\	\	\
No disengagement + conditions ~ (1|participant)	12	0	0.96	1	8.20	0.004	0.06

RD = residual deviance, AIC = Akaike information criterion, dAIC = difference between a model’s AIC and those of the best model, AICw = AIC weight, df = degrees of freedom of the chi-squared statistic, η^2^ = eta squared as the ratio between the chi-squared and the residual deviance of the null model.

**Table 3 ijerph-18-01592-t003:** Mean (and standard deviation) of the duration (ms), decibels (dB), and frequency (Hz) of strong syllables (S), weak syllables (W), and pure tones (PT) by the test conditions.

Conditions	Feature	Strong Mean (SD)	Weak Mean (SD)	Pure Tone
F	ms	409 (48)	278 (45)	
	dB	75 (5)	70 (5)	
	Hz	212 (34)	192 (31)	
NS	ms	412 (40)	296 (33)	412 (40)
	dB	75 (3)	68 (6)	75 (3)
	Hz	207 (17)	190 (24)	207 (17)
NW	ms	399 (45)	271 (18)	271 (18)
	dB	76 (3)	70 (3)	70 (3)
	Hz	206 (27)	192 (34)	192 (34)

**Table 4 ijerph-18-01592-t004:** Model comparison for predicting looking times during test phase conditions.

Model	RD	dAIC	AICw	df	χ	*p*	η^2^
Looking times ~ (1|participant)	42	22.5	0	\	\	\	\
Looking times ~ condition + (1|participant)	40	0	1	2	24.45	<0.001	0.036

## Data Availability

The data presented in this study are available in Appendix A, at the OSF repository link: https://osf.io/97r3e/?view_only=a7e02029afbc4908b9aaee0e9a358284.

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
