# Peer review of "An Integrated Perspective on Spatio-Temporal Attention and Infant Language Acquisition"

_ijerph, 2021, doi:10.3390/ijerph18041592_

Round 1

Reviewer 1 Report

  1. Were the families with similar socioeconomic status? How often the parents were with their infants daily? Did they speak to their infants in Italian the majority of the time every day? Did the authors know why the families were not able to participate in the second experiment? Did the families receive compensation for participating in the experiments? Were the families aware of the purpose of the study?
  2. How long did each experiment take? It would be helpful to include some literature regarding the attention span in infants. 
  3. What software did you use to create the audio stimuli in experiment 2?
  4. Were the stimuli in the NS NW conditions presented in random order?
  5. On page 10 line 399, did you mean NS=NS/(F+NS+NW)? 
  6. Where is the Appendix mentioned on Page 8 line 336?

Reviewer 2 Report

The present research paper studies the correlation of the ability for spatial disengagement with the ability for temporal disengagement and attention engagement in processing auditory language stimuli in infants. In this way, the work falls within the research line on early language acquisition and speech segmentation in young children, which is not new itself, but which still has some gaps in what concerns our knowledge of infant language development. In this respect, the application of the experimental protocol to Italian-learning children makes a nice contribution to the scientific understanding of language relativity in speech segmentation, although this question particularly is not explored in full in the paper.

My general evaluation of this paper proposal is positive, although there are several shortcomings I would like to comment.

  • I do not find the paper completely coherent in what concerns its theoretical background, on the one hand, and its results and conclusions, on the other. For example, in Introduction the authors make a question - “Where does this crucial ability to detect and benefit from such prosodic cues originate from?” – which is actually not answered by the results of the paper. The experimental part of the paper does not point at the origins of the ability to detect prosodic cues, but rather at its (possible) belonging to general cognitive control in speakers. Maybe, the authors should include more contextualizing information on how the initiation in detecting prosodic cues starts and what are the contextual and biological factors determining it. A brief overview of cognitive control and, specifically, attentional development after birth may be more suitable for the theoretical background of the present research.

  • The authors’ hypothesis is plausible, but I do not find it completely tested/analyzed in the paper. Experiment 1 includes both simple and complex stimuli, but there is no significant comment in either Results or Discussion on the effect of complex stimulus on auditory disengagement. Does it actually occur? Or is there no difference on temporal attention engagement from simple/complex performance? (this does not seem to be the case). In short, I would expect authors to comment in more detail on how the typology of spatial disengagement stimuli influence on further performance with temporal attention engagement.

  • Concerning the stimuli type, I miss authors’ comment on whether simple condition stimuli might not have developed familiarization among participants and, in this way, influence the arousal level. In this very line, I suggest authors to include information on whether they have designed all experiments themselves or whether they have replicated previous experimental designs, either fully or partially.

  • From a purely phonological point of view, I find objectionable the selection of phonological segments for Experiment 2. I find coherent the inclusion of voiced and voiceless plosives, and nasals, but I find it surprising that the experiment includes a fricative /v/ and, specifically, the lateral liquid /l/. Plosives and nasals are actually among the first consonant segments to be acquired, but fricatives and, particularly liquid phonemes (like /l/) are usually among the last ones in the phoneme order acquisition. Could the authors please argue why they chose /v/ and /l/? I would also encourage the author comment on the qualitative results from this experiment, that is, how specific consonant segments might have influenced the performance on the task.

  • Concerning experimental results, I find it surprising that the authors do not comment the impressive SD value which in some cases is almost as high as the result itself. Sample participants are extremely different in task performance and I believe to be a very important fact. Actually, it confirms for me that attentional development is extremely individual and that the obtained results just point at plausibly general tendency rather than to a specific strong correlation.

  • Finally, I find it necessary to comment the general discussion and the conclusions of this paper. The presented research does not point at any specific direction but rather confirms that language acquisition occurs even if attentional system is reduced, and this point should be clearly stated. Strong variability among participants is a good proof of it. In this respect, I would suggest the authors to extend their comment on attention effect on language acquisition comparing typical and atypical development. The results of the research itself refer to typical development only and, even more, they show that typically developing children may present reduced disengagement and attention engagement in comparison with other infants. What these results suggest is that, probably -and this should be experimentally testes- atypically developing children (not only SLE or autism, but also other chromosomic clinical pictures) would present even more reduced rated on similar tests.

Round 2

Reviewer 2 Report

I have read the new version of the paper and I believe authors to have significantly improved all the shortcomings mentioned in the 1st review. They have made the theoretical background more appropriate for contextualizing their research, new clarifying information has been included in sections on experiments and data analysis, and the paper discussion has very much improved in line with the research results.